# *FedTherapist*: Mental Health Monitoring with User-Generated Linguistic Expressions on Smartphones via Federated Learning

**Jaemin Shin**[1], **Hyungjun Yoon**[1], **Seungjoo Lee**[1], **Sungjoon Park**[2],
**Yunxin Liu**[3], **Jinho D. Choi**[4], **Sung-Ju Lee**[1]

[1]KAIST [2]SoftlyAI [3]Tsinghua University [4]Emory University

{jaemin.shin, hyungjun.yoon, seungjoo.lee, profsj}@kaist.ac.kr,
sungjoon.park@softly.ai, liuyunxin@air.tsinghua.edu.cn, jinho.choi@emory.edu

## Abstract

Psychiatrists diagnose mental disorders via the linguistic use of patients. Still, due to data privacy, existing passive mental health monitoring systems use alternative features such as activity, app usage, and location via mobile devices. We propose *FedTherapist*, a mobile mental health monitoring system that utilizes continuous speech and keyboard input in a privacy-preserving way via federated learning. We explore multiple model designs by comparing their performance and overhead for *FedTherapist* to overcome the complex nature of on-device language model training on smartphones. We further propose a Context-Aware Language Learning (*CALL*) methodology to effectively utilize smartphones' large and noisy text for mental health signal sensing. Our IRB-approved evaluation of the prediction of self-reported depression, stress, anxiety, and mood from 46 participants shows higher accuracy of *FedTherapist* compared with the performance with non-language features, achieving 0.15 AU-ROC improvement and 8.21% MAE reduction.

## 1 Introduction

Nearly a billion people worldwide are living with mental disorders, which seriously affect one's cognition, emotion regulation, and behavior. Early diagnosis and proper treatment are the keys to alleviating the negative impact of the mental disorder (Sharp and Lipsky, 2002). However, most patients have been unaware of their disorder for years (Epstein et al., 2010), which delays treatment while the symptoms worsen.

Given their ubiquity in users' lives, researchers have leveraged smartphones to resolve this unawareness problem, using features such as phone usage patterns, location, and activity for seamless mental health monitoring (LiKamWa et al., 2013; Wang et al., 2014, 2018; Li and Sano, 2020; Tlachac et al., 2022a). However, these features fail to reflect how licensed psychiatrists diagnose mental disorders by conversing with patients (Murphy et al., 2000). While analyzing linguistic use is ideal for monitoring smartphone users' mental health, substantial privacy concerns it raises present challenges in amassing sufficient data to train advanced NLP neural networks (Devlin et al., 2019).

We propose *FedTherapist*, a privacy-preserving mental health monitoring system that leverages user-generated text (speech and keyboard) on smartphones via Federated Learning (FL). FL decentralizes model training on client devices (e.g., smartphones) using locally stored data (McMahan et al., 2016), ensuring privacy on *FedTherapist* by only collecting securely aggregated model updates. For a detailed introduction to FL, see Appendix A.

Despite recent advances in FL + NLP (Lin et al., 2022; Xu et al., 2023; Zhang et al., 2023a), on-device training (i.e., training on smartphones) of language models for mental status monitoring remains unexplored. We explore the best model design for *FedTherapist* across multiple candidates, including Large Language Models (LLMs), by comparing their mental health monitoring performance and smartphone overhead.

In realizing *FedTherapist*, the challenge remains to effectively capture mental health-related signals from a large corpus of spoken and typed user language on smartphones, which differs from prior NLP mental health studies based on social media (Yates et al., 2017; Park et al., 2020) – see Appendix G. To address such a challenge, we propose *Context-Aware Language Learning* (*CALL*) methodology, which integrates various temporal contexts of users (e.g., time, location) captured on smartphones to enhance the model's ability to sense mental health signals from the text data. Our evaluation of 46 participants shows that *FedTherapist* with *CALL* achieves more accurate mental health prediction than the model trained with non-text data (Wang et al., 2018), achieving 0.15 AUROC improvement and 8.21% reduction in MAE.

## 2 Data Collection

We conducted an IRB (Institutional Review Board)-approved data collection study to evaluate *FedTherapist* on real-world user data. Although FL works without user data collection, we collected the data to fully explore the potential of smartphone text data on mental health monitoring. We recruited 52 participants over the Amazon Mechanical Turk (MTurk) who identified English as the first and only language they use daily. Participants collected data for 10 days on our Android application, where we provided no instructions or restrictions to the participants' behavior during the study so that we collect data from their normal daily routine. The study details are in Appendix B, including how we assured the integrity of our data, removing six abnormal participants from the evaluation.

We collected two types of text input: (1) *speech*: user-spoken words on the microphone-equipped smartphones, and (2) *keyboard*: typed characters on smartphone keyboards. On average, we collected $13{,}492\pm14{,}222$ speech and $4{,}521\pm5{,}994$ keyboard words per participant. We describe how we implemented a system to collect these inputs on smartphones in Appendix C. To compare *FedTherapist* with the non-text data, we collected the data used in a mobile depression detection study (Wang et al., 2018) detailed in Section 4.1.1, along with the text data. We collected ground truth on four mental health statuses: depression, stress, anxiety, and mood, as detailed in Section 4.1.2.

To ensure participant privacy, we carefully controlled data collection, storage, and access as in the Ethics Statement. Given the highly sensitive nature of the data, it will not be released publicly and will be deleted post-study following IRB guidelines.

## 3 *FedTherapist*

### 3.1 Mental Health Prediction Model

*FedTherapist* aims to deliver accurate mental health monitoring using user-generated text data from smartphones, employing FL. Given that FL necessitates training a model on resource-constrained user smartphones, it is essential that our model remains accurate and lightweight. We explore three candidate model designs: (1) *Fixed-BERT + MLP*: The model consists of a text embedding model followed by a multilayer perceptron (MLP). We selected BERT (Devlin et al., 2019) as our text embedding method for its effectiveness in docu-

| Methods | Performance | Smartphone Overhead* | |
| --- | --- | --- | --- |
| | Depression[†] | CPU[‡] | Memory |
| Fixed-BERT + MLP | 0.716 | 35% | 219MB |
| End-to-End BERT + MLP | 0.524 | 68% | 864MB |
| LLM | 0.406 | N/A[§] | N/A[§] |

Table 1: Comparison of candidate mental health prediction methods. *: averaged measurement on Google Pixel (2016) and Samsung Galaxy S21 (2021). [†]: measured in AUROC; [‡]: measured in average per-core utilization; [§]: failed loading on test smartphones.

ment classification tasks (Adhikari et al., 2019). We adopt pre-trained BERT and only train MLP via FL. (2) *End-to-End BERT + MLP*: We use the same model architecture as the former but perform end-to-end training, including BERT and MLP. (3) *LLM*: Given recent breakthroughs and state-of-the-art performance of Large Language Models (LLMs) (OpenAI, 2023), we train and use LLM to prompt user text and output mental health status.

We compared the model designs by measuring the depression detection performance (Section 4.1.2) and their feasibility of operation on two smartphones as noted in Table 1. We adopted widely used base BERT models such as Distil-BERT (Sanh et al., 2019) and RoBERTa (Liu et al., 2019) and reported the best performance. We used LLaMa-7B (Touvron et al., 2023) as LLM, one of the smallest publicly available LLM, for its feasibility on smartphones. We applied LoRA (Hu et al., 2022) for *End-to-End BERT + MLP* and *LLM* experiments, for lightweight fine-tuning at resource-constrained mobile scenarios. We conducted experiments using Leave-One-User-Out (LOUO) cross-validation, segmenting the users into 1 test user, 5 for validation, and 40 for training; the experiments were repeated 46 times by setting each user as a test user, with the subsequent five users serving as the validation set. Appendix D and E.4 present more details on experimental methods.

Table 1 shows the comparison results. We found that inference and training of *Fixed-BERT + MLP* and *End-to-End BERT + MLP* are both available on our test smartphones, while *LLM* failed due to its memory constraints, requiring $> 8$GB free memory to load 8-bit quantized LLaMa-7B. *Fixed-BERT + MLP* results in 0.192 and 0.310 higher AUROC than *End-to-End BERT + MLP* and *LLM*, respectively, in depression detection, while it incurs less smartphone overhead by training only MLP. We hypothesize that the subpar performance of BERT and LLM results from overfitting on our 40-user training sample, as shown in Figure 6. Past

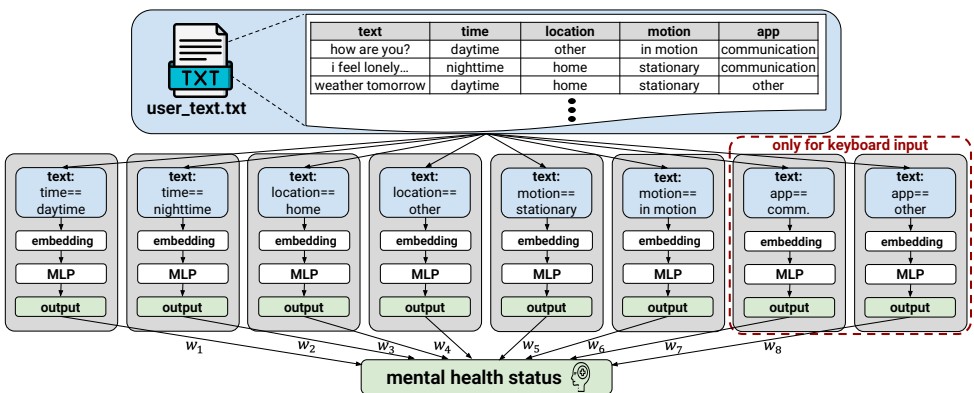

Figure 1: Design of Context-Aware Language Learning (*CALL*) methodology for *FedTherapist*. *CALL* maps the user's text to multiple temporal contexts (time, location, motion, and app). Each piece of text trains the relevant models on each context; for example, the text generated at *daytime* and *home* is used to train both the 'daytime' and 'home' models. *CALL* ensembles the context model outputs to determine the mental health status of a user.

research (Dai and Le, 2015; Sun et al., 2019) corroborates that pre-trained models can overfit on small-scale datasets, compromising their generalizability. This could be mitigated by using a larger training dataset from broader population.

*Fixed-BERT + MLP* is also superior in privacy, as partially training the model addresses concerns that gradient sharing during FL could result in data leakage or even sentence reconstruction (Huang et al., 2021; Gupta et al., 2022). While we acknowledge the risk of attackers possibly reconstructing original text after recovering the embedded text representations (Coavoux et al., 2018; Song and Raghunathan, 2020), we could further integrate protection methodologies to such an attack (Lee et al., 2022). Thus, we use *Fixed-BERT + MLP* as our model design for *FedTherapist*, with Distil-BERT that showed the best performance (Table 3).

## 3.2 Context-Aware Language Learning

As an average person speaks more than 16,000 words and sends 71 text messages per day (Mehl et al., 2007; Harrison and Gilmore, 2012), processing such a large corpus of noisy text input on smartphones for accurate mental health sensing is yet unexplored. To address such a challenge, we propose *Context-Aware Language Learning (CALL)* methodology for *FedTherapist*.

We designed *CALL* to integrate various temporal contexts of a user (e.g., time, location, etc.) captured on a smartphone to enhance the model's ability to sense mental health signals from the text data. The intuition behind our design is to let the neural network focus on the user-generated text in a context in which users are highly likely to expose their mental health status. For example, users would be more likely to express their emotions via

text at night at home than daytime at the workplace.

We integrate four types of contexts that are available on smartphones, each based on previous psychological studies as follows:

- *Time (T)*: Time, indicative of when text generation happens, was chosen as context due to humans' daily emotional cycles (English and Carstensen, 2014; Trampe et al., 2015). These studies imply significant emotional differences between morning and evening, hence we divide 24 hours into two contexts: daytime ($T_D$: 9AM~6PM) and nighttime ($T_N$: 6PM~9AM).
- *Location (L)*: Smartphone GPS infers user location during text generation. Motivated by a study (Lively and Powell, 2006) showing emotional expression is influenced by the domain (home or workplace), we divided GPS data into two contexts: home ($L_H$) and other locations ($L_O$). We identified the most frequented location from clustered GPS data as home.
- *Motion (M)*: Smartphone accelerometers indicate user movement during text generation. Inspired by a study (Gross et al., 2012) revealing emotion expression varies with physical activity, we used Google Activity Recognition API (Google, 2022a) to distinguish two contexts: stationary ($M_S$) and in motion ($M_M$).
- *Application (A)*: We categorized applications used for typing into two contexts: communication ($A_C$) and others ($A_O$), based on studies predicting mental status from text messages (Tlachac et al., 2021, 2022c). Applications in the 'Communication' category of Google Play Store, such as WhatsApp, are classified as communication; otherwise classified as others.

Figure 1 depicts the mental health prediction model of *CALL*, which aggregates the user text on

| Methods | CL+NonText | FL+Text | | | *FedTherapist* (FL+Text+*CALL*) | | |
|---|---|---|---|---|---|---|---|
| Data Type | Non Text Data | Speech (S) | Keyboard (K) | S+K | Speech (S) | Keyboard (K) | S+K |
| Depression (AUROC ↑) | 0.625±0.010 | 0.627±0.004 | 0.710±0.007 | **0.775±0.010** | 0.571±0.003 | 0.746±0.000 | 0.721±0.008 |
| Stress (MAE ↓) | 20.83±0.03 | 23.44±0.02 | 24.21±0.05 | 24.78±0.03 | 21.34±0.01 | 20.07±0.01 | **19.12±0.01** |
| Anxiety (MAE ↓) | 20.95±0.06 | 25.80±0.03 | 26.85±0.08 | 27.30±0.03 | 22.56±0.05 | 21.39±0.01 | **20.56±0.02** |
| Mood (MAE ↓) | 18.76±0.09 | 22.85±0.03 | 23.21±0.02 | 23.67±0.02 | 19.11±0.02 | 19.18±0.00 | **18.57±0.02** |

Table 2: Mental health monitoring performance on different methods.

each temporal context and trains separate models. *CALL* performs ensemble learning on $N$ context models and takes a weighted sum on the models' outputs to determine the user's mental health status. We use $N = 8$ for keyboard input with two contexts each from $T$, $L$, $M$, $A$ contexts and $N = 6$ for speech input without $A$ contexts. When we use both speech and keyboard input, we use all the contexts and use $N = 14$. We implemented two types of ensemble weights in our experiments: (1) $E_A$: averaging the model outputs ($w_1$, $w_2$, ..., $w_N$ = $\frac{1}{N}$); and (2) $E_E$: weighted sum of model outputs using the trained ensemble weights $W$ over FL.

In terms of privacy, an attacker could reveal the private information of a user based on which context models' gradients are non-zero. For example, if a user's $L_H$ model gradient is non-zero while the $L_O$ gradient is zero, an attacker could infer that a user stayed home most of the time. To address this problem, we could adopt Secure Aggregation protocol (Bonawitz et al., 2017) that securely aggregates the FL clients' model gradients to prevent attackers from acquiring individual model updates.

# 4 Evaluation

We answer the following key questions: (1) How beneficial is using the text data on mental health monitoring tasks? (2) How much performance improvement does *FedTherapist* achieve with *CALL*? (3) How do each context model and ensemble methods of *CALL* perform?

## 4.1 Experimental Setup

### 4.1.1 Methods and Data Types.

We compare three methods: (1) *CL+NonText*: we use Centralized Learning (CL) to train an MLP model on the non-text data utilized by a previous study (Wang et al., 2018), which uses the following features: stationary time, conversation count, sleep end time, location duration, unlock duration, unlock counts, and number of places visited. We extracted the features by preprocessing the device logs and raw sensor values such as GPS, accelerometer, or ambient light sensor from participant's mo-

bile devices. (2) *FL+Text*: we use Federated Learning (FL) to train a *Fixed-BERT + MLP* model with the speech and keyboard input text data captured on smartphones collected from our data collection. While FL is often outperformed by CL (Nilsson et al., 2018), CL on text data is avoided due to privacy concerns. (3) *FedTherapist*: we applied *CALL* to *FL+Text*. We tested three sets of text data; speech input only (S), keyboard input only (K), and both (S + K). We employed the Leave-One-User-Out (LOUO) cross-validation technique in our experiments. Out of 46 users, one was configured as the test user and the remaining 45 were used for training. This setup was iteratively conducted 46 times, setting each user as the test user. We report the test performance after running 1,000 epochs for CL or rounds for FL. Detailed experimental methodologies are presented in Appendix D.

### 4.1.2 Tasks and Metrics.

We evaluated on following tasks and metrics: (1) *Depression (Classification)*: post the 10-day study, depression was surveyed on participants through the PHQ-9 test questionnaire (Wang et al., 2018). We label a participant as mildly depressed if PHQ-9 score $\geq 5$ (Kroenke and Spitzer, 2002), and use the 10-day study data for prediction. We test the Area Under the ROC curve (AUROC) as a metric. (2) *Stress, Anxiety, and Mood (Regression)*: we use the data from each day to predict the level of stress, anxiety, and mood, which were self-rated by participants on a 0∼100 scale as in previous literature (Li and Sano, 2020); 0 and 100 each indicate the negative and positive feelings. We test the Mean Absolute Error (MAE) as a metric to evaluate regression.

## 4.2 Comparison of Data Types and Methods

Table 2 displays the performance of three methods for mental health monitoring. In the depression task, which utilizes 10-day data input, *FL+Text* outperforms *CL+NonText* by 0.15 AUROC. Conversely, *CL+NonText* excels over *FL+Text* in remaining tasks that use one-day data input, with 2.61∼6.35 lower MAE. These results suggest that

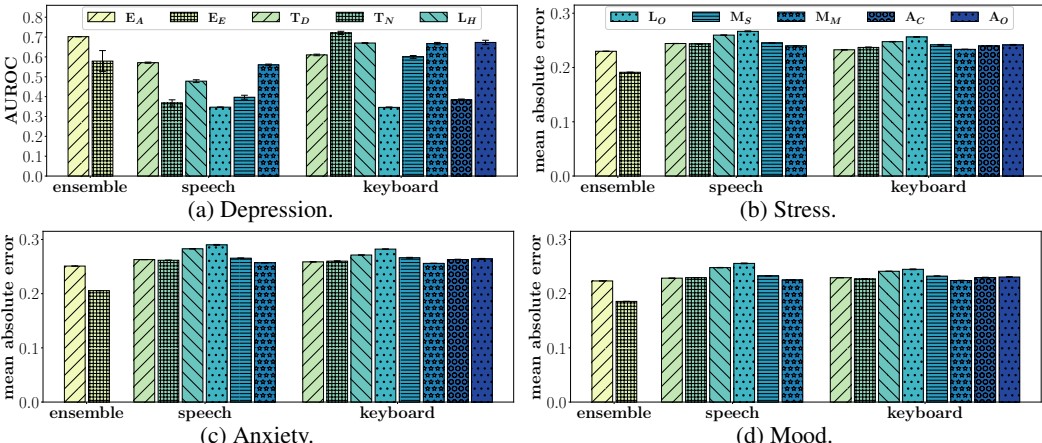

Figure 2: Performance of ensemble methods ($E_A$: ensemble by averaging, $E_E$: ensemble by weighted sum) of *CALL* and context models ($T_D$: daytime, $T_N$: nighttime, $L_H$: home, $L_O$: other locations, $M_S$: stationary, $M_M$: in motion, $A_C$: communication applications, and $A_O$: other applications) on four mental health monitoring tasks. All subgraphs share the same legend with Figure 2a and 2b.

longer-duration text input more effectively captures mental health signals, likely due to the complexity of detecting these signals within the noisy text, thus leading models to struggle with shorter-term data.

Unlike *FL+Text*, we find that *FedTherapist* consistently achieves improved performance over *CL+NonText* on all tasks, achieving 0.121 higher AUROC on *depression* and 0.19~1.71 lower MAE on remaining tasks. *FedTherapist* also achieves enhanced performance on *stress*, *anxiety*, and *mood* tasks than *FL+Text* with 4.32~6.74 lower MAE, with comparable AUROC (0.746 vs 0.775) on *depression*. These results demonstrate that *FedTherapist* with *CALL* generally improves our model in detecting mental health signals from the raw text, regardless of the data collection period.

Our findings recommend the combined use of speech and keyboard input (S+K) with *FedTherapist*, owing to its superior results in stress, anxiety, and mood tasks (achieving a lower MAE between 0.54 and 2.22) and a similar performance in depression tasks (0.721 vs. 0.746) over single text type approaches (S or K). S and K each excels at different tasks; S outperforms in *mood* with *FedTherapist*, while K shows superior performance in other tasks.

### 4.3 Comparison of Context Models and Ensemble Methods of *CALL*

Figure 2 illustrates the performance of each context model and ensemble methods of *CALL* (Section 3.2) across four mental health monitoring tasks (Section 4.1.2). Among ensemble methods, the averaging method $E_A$ excels on the *depression* task with 0.702 AUROC but underperforms elsewhere. The weighted ensemble, $E_E$, records the lowest

MAE for remaining *stress* (19.12), *anxiety* (20.56), and *mood* (18.57) tasks, albeit with an AUROC of 0.579 for *depression*. This implies that some context models are not always helpful for *stress*, *anxiety*, and *mood* tasks, that weighted sum outperforms the averaging, unlike the *depression* task.

Among the time contexts, $T_N$ performs better on the *depression* task on the keyboard, while $T_D$ excels in speech, indicating more emotional expression at daytime speech and nighttime keyboard use. For the location, $L_H$ outperforms $L_O$ in both speech and keyboard, suggesting greater mental state disclosure at home. With the motion contexts, texts during movement ($M_S$) provide superior mental state indicators over $M_M$. For the application contexts, $A_O$ significantly improves AUROC over $A_C$, underscoring the value of considering non-communication app inputs, unlike prior studies that focus only on the messenger app inputs (Tlachac and Rundensteiner, 2020; Tlachac et al., 2022c).

## 5 Conclusion

We presented *FedTherapist*, a mobile mental health monitoring system leveraging continuous speech and keyboard input while preserving user privacy via Federated Learning (FL). Our Context-Aware Language Learning (*CALL*) empowers *FedTherapist* to effectively sense the mental health signal from a large and noisy text data on smartphones. Our evaluation with 46 participants on depression, stress, anxiety, and mood prediction shows that *FedTherapist* outperforms the model trained on non-language features. We believe *FedTherapist* demonstrated the feasiblity of on-device NLP on smartphones that enable usage of user-generated linguistic data without privacy concerns.

## Limitations

Unlike previous approaches (Wang et al., 2014, 2018; Li and Sano, 2020) on mobile mental health monitoring that collected data over 10 weeks, our data collection was held for only 10 days. Moreover, our study pool of 46 English-speaking participants do not represent regional & cultural differences of language among users. Thus, the reported evaluations should be considered as exploratory. Note that our study gathered participants from US and Canada, spanning ages from 20s to 60s, while previous studies were conducted mostly on university students. Studying *FedTherapist* on different languages for a longer term would be the next step for generalizing our findings.

The accuracy of our speech collection in unconstrained environments was not investigated. To avoid collecting the voice of non-target users and only record the speech in high quality, our *speech text collection module* (Section C.1) integrated Voice Activity Detector (Google, 2022b), VoiceFilter (Wang et al., 2019, 2020), a language classifier (Team, 2021), and a speech recognition model (Cephei, 2022). Such machine learning-driven components, especially VoiceFilter, may yield errors at noisy or untrained environments (Eskimez et al., 2022). However, the speech data in our experiment shows comparable or better results on *stress*, *anxiety*, and *mood* tasks with accurately collected keyboard data; the speech data also improves performance when it is jointly used with the keyboard data with *FedTherapist*. This result hints that the speech data collected from our *speech text collection module* could be effectively utilized in mental health monitoring tasks.

When comparing the model designs for *FedTherapist* in Section 3.1, we were limited to train on larger batch sizes (e.g., 128) for BERT and LLM as indicated in Appendix D.2, due to our GPU memory restrictions (24GB on NVIDIA RTX 3090). Thus, our findings on comparison between model designs could be altered when tested on more powerful hardware or different data, but we still believe *Fixed-BERT + MLP* is a promising option for mobile scenarios, given its low system overhead and high mental health monitoring performance.

## Ethics Statement

The collection of text data on user smartphones has been previously discouraged as it contains highly private information (Tlachac et al., 2022a). As the privacy threat on participants should always be carefully considered (Šuster et al., 2017; Benton et al., 2017), we performed the following based on a previous work (Chen et al., 2019) to minimize the risk that originates from our study. (1) We provided detailed instructions on all data types we are collecting during the study to the participants. (2) We only accepted participants who provided their own signatures to confirm that they are aware of the types of collected data. (3) We used VoiceFilter (Wang et al., 2019), the state-of-the-art voice separation technology to avoid collecting the speech of others who are not part of our study. (4) The collected data on smartphones were saved only in our Android application's sandbox storage where access from other applications is restricted. (5) The data was securely transmitted through encrypted communication with our SSL-certified and HIPAA-compliant database server. (6) The data was available only to the authors from the same institution as the corresponding author. (7) We do not plan to share the collected data externally and will delete the data when our IRB approval expires.

## Acknowledgments

This work was supported by the Institute of Information & communications Technology Planning & Evaluation (IITP) grant funded by the Korea government (MSIT) (No. 2022-0-00064) and the Institute of Information & communications Technology Planning & Evaluation (IITP) grant funded by the Korea government (MSIT) (No. 2022-0-00495).

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

## A   Federated Learning

Federated Learning (FL) (McMahan et al., 2016; Li et al., 2020; Shin et al., 2022; Liu et al., 2022) is a recent machine learning paradigm that enables model training without data collection. FL globally trains a model from the distributed user data on multiple mobile devices. We apply FL with *FedTherapist* to utilize the privacy-sensitive text data on smartphones for effective mental health monitoring. The most commonly adopted FL methodology is FedAvg (McMahan et al., 2016), which operates synchronously across clients over multiple rounds as follows: at the beginning of each round, the FL server randomly samples $K$ clients out of total $N$ clients ($K \ll N$). The sampled clients download the up-to-date model weights $w_R$ at round $R$ then train on the local client data for $E$ epochs. The client's updated model ($w_{R+1}^i$ for client $i$) gets uploaded to the server, which is further aggregated into a new updated model as $w_{R+1} \leftarrow \sum_{i=1}^{K} \frac{n_i}{n} w_{R+1}^i$, where $n_i$ indicates client $i$'s data size and $n$ indicates the data size sum on $K$ sampled clients.

Various recent healthcare applications, such as brain cancer imaging (Sheller et al., 2020; Lee et al., 2021), clinical support for COVID-19 (Dayan et al., 2021; Vaid et al., 2021), MRI data analysis (Silva et al., 2019), blood pressure monitoring (Brophy et al., 2021), Parkinson's diseases diagnosis (Chen et al., 2020), and stress level prediction (Can and Ersoy, 2021) have employed FL. Additionally, FL has been recently approached on NLP tasks, such as next word prediction (Xu et al., 2023), semantic parsing (Zhang et al., 2023a), and legal text data processing (Zhang et al., 2023b). None of the previous approaches, however, capitalize a user's speech and keyboard input on smartphones with FL. We propose *FedTherapist*, a mental health monitoring system to leverage user-generated linguistic expressions on smartphones with FL.

## B   Data Collection Study Details

The data collection study was conducted for three weeks. Online Amazon Mechanical Turk (MTurk) participants freely joined our study during that period by downloading our Android application, completing a survey, and executing three tasks as illustrated in Figure 3. The survey asked the participants to confirm if they were fully aware of the data that were being collected during the study. The data collection study for each user lasted for ten days, with

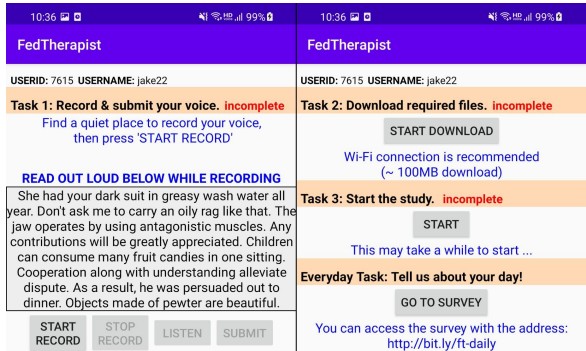

Figure 3: Screenshots of our data collection application. Participants completed three tasks to join the study. First, participants were asked to upload their voice sample (Task 1) to use VoiceFilter (Wang et al., 2019) (Section C.1). Participants then downloaded the model files used in our local data collection module (Task 2), and started the study (Task 3). We asked the participants for a daily mental health status report during the study.

$72 compensation. Figure 4 shows the word count of collected data on each participant, where an average of 13,492±14,222 and 4,521±5,994 words are collected on speech and keyboard input, respectively. Figure 5 shows the count of the participant responses on each mental health status.

As we conducted our study via MTurk, we could not directly monitor whether users fully followed our instructions while collecting data. For example, we could not confirm if users installed our data collection application on their main smartphone. After collecting data, we removed six out of 52 participants from the evaluation for showing abnormal data statistics. Three were excluded for not having keyboard input throughout the data collection duration. The other three were dropped for being completely stationary for ten days, which could imply that the users might have installed the application on their secondary phones.

## C   Data Collection App Implementation

### C.1   Speech Text Collection Module

Capturing user speech from the continuous audio input on a smartphone presents several challenges: (1) As the user could speak at any time of the day, continuous microphone recording is desired, leading to a significant smartphone battery drain. (2) The voice of non-target users could be included in the audio. This must be excluded not only to achieve effective mental health prediction of the target user but also to protect the privacy of non-

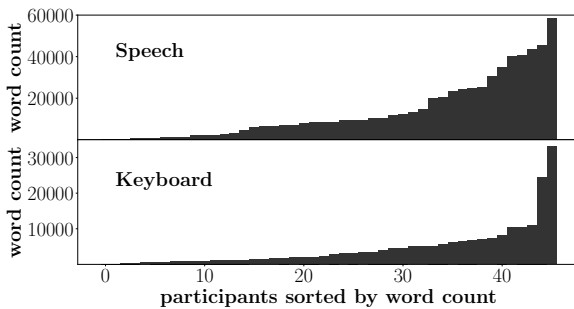

Figure 4: Word count of each participant's collected speech and keyboard input from the 10-day study.

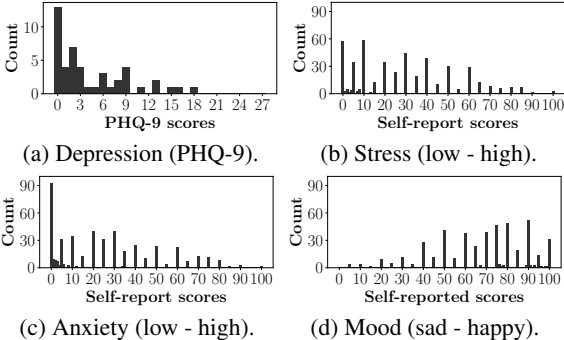

(a) Depression (PHQ-9).

(b) Stress (low - high).

(c) Anxiety (low - high).

(d) Mood (sad - happy).

Figure 5: Count of participant responses on different mental health status scores.

target users. (3) The user voice could be recorded in low quality with noise, as the recording occurs in unconstrained, real-world environments. The distance between the user's mouth and the smartphone microphone could be far, and the background noise in the audio makes it difficult to acquire a clear voice from a user.

We implemented our *speech text collection module* to collect the user speech data while mitigating the aforementioned challenges as follows: To avoid continuous recording (Challenge 1), we adopted a *duty cycling* system with a Voice Activity Detector (VAD) model (Wang et al., 2014): it starts recording and infers on a VAD model for a minute in every four minutes to check whether there is an ongoing conversation. The module continues recording if a conversation is detected, where the recorded audio is inferred on VoiceFilter (Wang et al., 2019, 2020), a model that removes the non-target users' voice based on the target user's sample voice (Challenge 2). Next, the VAD model is applied to the VoiceFilter output to remove audio without the target user's voice. To ensure that the processed audio contains the target user's voice (Challenge 3), we designed the module to leverage a language classifier that outputs the language of a

given audio file. If the classifier outputs a language that the user speaks, we regard the audio as high-quality recording of the user's voice and apply a speech recognition model to retrieve the text data.

We implemented the *speech text collection module* on our Android app to demonstrate its on-device capability and conduct *FedTherapist* evaluation on real-world user devices. We adopted three publicly available models: a lightweight VAD model from Google WebRTC (Google, 2022b), a language classifier from Silero (Team, 2021) that searches among 95 languages, and an offline speech recognition model, Vosk (Cephei, 2022) (en-us model). To enable inference on smartphones, we reduced the number of convolution layers from eight to four and changed the bidirectional LSTM to uni-directional LSTM on the VoiceFilter model. For models that do not provide an Android implementation (e.g., VoiceFilter), we used MNN (Jiang et al., 2020) to convert the PyTorch version of the models into an Android executable. To minimize the smartphone battery drain and preserve the user experience, we implemented to fully execute the module only when the device is idle and being charged. The recorded audio files from the duty cycling system are temporarily saved on the device until the device gets connected to a charger.

### C.2 Keyboard Text Collection Module

The *keyboard text collection module* captures all the input text the user enters with their smartphone keyboard. Our implementation on our app tracked the text input events on Android smartphones using the Android Accessibility Service (Android, 2021). We preprocessed the collected text as follows (Uysal and Gunal, 2014; Vijayarani et al., 2015): (1) Emails, hashtags, links, mentions, punctuations, and numerical values were removed; (2) emojis were replaced with CLDR (Common Locale Data Repository) Short Names (Unicode, 2022), and abbreviations were replaced with what they stand for; (3) the letters that were excessively repeated were reduced; and (4) typos were corrected using the auto-correction API (Norvig, 2016).

### D Detailed Experimental Methodologies

### D.1 Handling Long Input Sequences

Our model input for four mental health prediction tasks (Section 4.1.2) often exceeds the max input size of the models, such as BERT (512 words on BERT-base (Devlin et al., 2019)) or LLM (2048

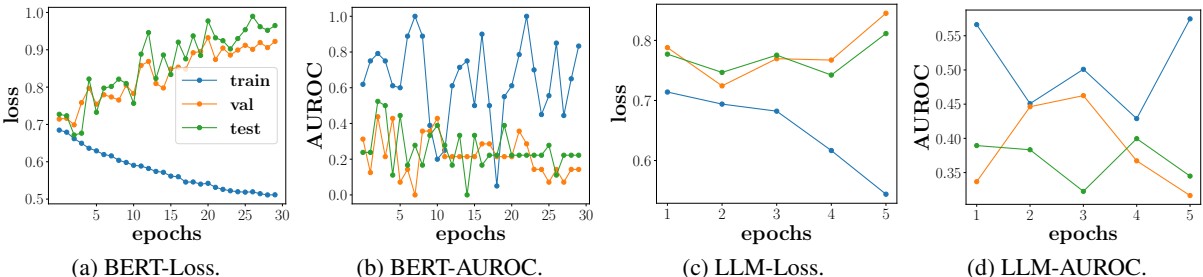

| (a) BERT-Loss. | (b) BERT-AUROC. | (c) LLM-Loss. | (d) LLM-AUROC. |

Figure 6: Loss and AUROC results on train, validation, and test samples over epochs from fine-tuning experiments in Section 3.1. Figure 6a and 6b indicate fine-tuning experiment of *End-to-End BERT + MLP* using a pre-trained RoBERTa-base (Liu et al., 2019) model with a learning rate of 0.0001. Figure 6c and 6d depict fine-tuning experiment of *LLM* on a pre-trained LLaMa-7B model with a learning rate of 0.0002. More details on experimental methods are found in Section 3.1 and Appendix D. All subgraphs share the same legend with Figure 6a.

words on LLaMa-7B (Touvron et al., 2023)). In such cases, we divided the input text into 512-word chunks and tested the following methodologies, inspired by Sun et al. (2019): (1) *full+pool*: we forward the chunks through the embedding model and then a maxpool layer. One embedding vector is generated, which we further use for prediction with multilayer perceptron (MLP). We include as many chunks as our GPU memory permits. (2) *single*: we forward chunks individually through the embedding model and MLP. We test the model performance by predicting based on the averaged logits from multiple chunks. We experimented with both options and reported better performance.

## D.2 Training and Cross Validation

For Section 3.1 experiments with training BERT and LLM, to target resource-constrained mobile scenarios, we applied LoRA (Hu et al., 2022), a lightweight fine-tuning technique. Moreover, we trained the 8-bit quantized LLaMa-7B model as LLM for resource-efficient training. For the BERT experiment, we applied DistilBERT-base (Sanh et al., 2019), RoBERTa-base (Liu et al., 2019), and BERT-base (Devlin et al., 2019) and reported the best performance. For each experiment, we applied sequence classification models and LoRA from PEFT (Parameter-Efficient Fine-Tuning) library (Mangrulkar et al., 2022), both from HuggingFace (Wolf et al., 2020). Based on Hu et al. (2022), we adopted the learning rate from a set of [0.0005, 0.0004, 0.0003, 0.0002, 0.00003] with a linear learning rate decay scheduler. We fine-tuned BERT and LLM until 30 and 5 epochs, respectively, and reported the test AUROC from an epoch with minimum validation loss. Given our GPU memory

restrictions, we selected a batch size of 16 for *single* experiments and 2 for *full+pool* experiments.

For Section 4 experiments with FL (*FL+Text* and *FedTherapist*), we applied FedAvg (McMahan et al., 2016) (Appendix A), a widely-used FL algorithm; exploring other FL approaches is not the focus on this work. We report better performance among the context models and ensemble methods for *CALL* experiments. We trained a Logistic Regression (LR) model as an MLP and applied Lasso regularization for the regression. A learning rate of 0.01 and a batch size of 10 were adopted.

For cross-validation, we employed the Leave-One-User-Out (LOUO), splitting participants into a single test user and training users. We trained the model on the training users for 1000 epochs for CL and 1000 rounds for FL. We then test on the single test user, with each participant rotated as the test user. We used three random seeds to repeat and average the performance.

## E Evaluation Results

### E.1 Comparison with Existing Methods

We conducted a comprehensive comparative analysis of *FedTherapist* against closely related approaches:

- *CNN + MLP*: Modeled after Yates et al. (2017) and Kim (2014), this CNN-based text classification employs a convolutional layer and a MLP. We used Yates et al. (2017)'s hyperparameters for depression identification.

- *Bag-of-words + MLP*: This model employs an MLP with a bag-of-words input with TF-IDF, based on Pirina and Çöltekin (2018)'s depression detection. We experimented with a n-gram range

of [1, 3] and representation sizes from [1000, 5000, 10000, 20000].

- *Empath + MLP*: Taking cues from Tlachac et al. (2021), which utilizes smartphone SMS messages, the model utilizes output representations from Empath (Fast et al., 2016), which analyzes text over 200 lexical categories.

To ensure a fair comparison, we trained all baseline models using learning rates from [0.0001, 0.001, 0.01, 0.1] and evaluated their performance at each of the 1000 epochs. We measured the best performance achieved by these models over the 1,000 epochs. For *FedTherapist*, we reported the test AUROC after its 1,000 epochs without selecting its peak performance. Despite this, *FedTherapist* still demonstrated superior results of 0.721 AUROC, while *CNN + MLP*, *Bag-of-words + MLP*, and *Empath + MLP* each resulted in 0.418, 0.564, 0.635 AUROC respectively.

The result shows that *FedTherapist* outperforms other methods in depression detection on our dataset. Limited training samples from only 46 users in our dataset might have hindered the effectiveness of CNNs trained from scratch — in contrast, Yates et al. (2017) trained with Reddit posts from more than 100,000 users. Other results suggest that using frozen BERT in *FedTherapist* shows better mental health monitoring performance than using bag-of-words or Empath.

### E.2    Comparison of Text Embedding Models

Table 3 reports the performance of the following text embedding methods: widely used BERT models (BERT-base, BERT-large (Devlin et al., 2019), RoBERTa (Liu et al., 2019)), lightweight BERT models tailored for mobile devices (ALBERT (Lan et al., 2020), MobileBERT (Sun et al., 2020), DistilBERT (Sanh et al., 2019)), BART base (Lewis et al., 2020), and a text embedding model for longer sequences, BigBird (Zaheer et al., 2020). For each model, we used pretrained uncased models from HuggingFace (Wolf et al., 2020), and used the *full+pool* method (Appendix D.1) for user texts exceeding a model's maximum input length.

DistilBERT outperforms other methods in *depression*, while MobileBERT outperforms other methods in *stress*, *anxiety*, and *mood*. As MobileBERT shows the lowest performance on *depression*, we selected DistilBERT for our *FedTherapist* evaluation in Section 4, which performs comparably to other methods in other tasks. BERT with

a larger parameter size (BERT-base, BERT-large) or a model for longer input sequences (BigBird) underperforms the lighter models on all tasks. We suspect that these disparities between models arise from variations in their pretraining data and methods, particularly in how well they correlate with our user-generated speech and keyboard data.

### E.3    Fine-tuning BERT for *FedTherapist*

In our *Fixed-BERT + MLP* model for *FedTherapist*, we utilize pretrained BERT models trained on traditional text corpuses such as Wikipedia or BookCorpus (Devlin et al., 2019). As *FedTherapist* primarily deals with conversation-based, user-generated text data from smartphones, we evaluated unsupervised fine-tuning of BERT on SODA (Kim et al., 2022), a large-scale dialogue dataset.

Three pretrained models, BERT-base, RoBERTa-base, and DistilBERT-base, were used. BERT-base was fine-tuned using Masked LM (MLM) and Next Sentence Prediction (NSP) objectives. We utilized only the MLM objective for RoBERTa-base, mirroring RoBERTa-base's strategy (Liu et al., 2019). For DistilBERT-base, we distilled the knowledge from our SODA-pretrained BERT-base model. We conducted fine-tuning in two ways: (1) *SODA*: we consider each dialogue in the SODA dataset as a single input and (2) *SODA-single*: the dialogue was divided per speaker and trained as separate inputs, reflecting *FedTherapist*'s input data that only contains texts from a single mobile user.

Table 4 reveals that fine-tuning with BERT-base significantly improved performance (0.197 AUROC) on the *depression* task, while RoBERTa-base and DistilBERT-base showed comparable performance ($\sim 0.7$ AUROC). *SODA* and *SODA-single* resulted in similar performances, potentially due to a lack of modeling objectives to understand conversation structure. We leave the inclusion of such objectives as future work.

### E.4    System Overhead of *FedTherapist*

We tested the system overhead of *FedTherapist* on two scenarios: (1) data collection and (2) on-device training. For the data collection scenario, we tested when the device is performing (i) duty cycling and (ii) full execution of the *speech text collection module* (Appendix C.1), which runs only when the device is idle and being charged. Note that we measured only the speech data collection and not keyboard data collection as the latter is lightweight without running any deep learning models.

| Methods | ALBERT-base | MobileBERT | DistilBERT-base | RoBERTa-base | BERT-base | BERT-large | BART-base | BigBird |
| Parameter Size | 11M | 25M | 66M | 110M | 110M | 340M | 140M | 128M |
|---|---|---|---|---|---|---|---|---|
| Depression (AUROC ↑) | $0.526_{\pm 0.007}$ | $0.319_{\pm 0.003}$ | $0.775_{\pm 0.010}$ | $0.729_{\pm 0.002}$ | $0.560_{\pm 0.011}$ | $0.622_{\pm 0.004}$ | $0.636_{\pm 0.005}$ | $0.626_{\pm 0.009}$ |
| Stress (MAE ↓) | $24.75_{\pm 0.02}$ | $23.98_{\pm 0.01}$ | $24.78_{\pm 0.03}$ | $24.59_{\pm 0.05}$ | $24.80_{\pm 0.06}$ | $24.76_{\pm 0.04}$ | $24.29_{\pm 0.05}$ | $24.95_{\pm 0.05}$ |
| Anxiety (MAE ↓) | $26.95_{\pm 0.02}$ | $25.99_{\pm 0.01}$ | $27.30_{\pm 0.03}$ | $27.05_{\pm 0.06}$ | $27.07_{\pm 0.04}$ | $27.05_{\pm 0.03}$ | $26.74_{\pm 0.04}$ | $27.25_{\pm 0.04}$ |
| Mood (MAE ↓) | $23.57_{\pm 0.02}$ | $22.82_{\pm 0.01}$ | $23.67_{\pm 0.02}$ | $23.79_{\pm 0.02}$ | $23.54_{\pm 0.02}$ | $23.01_{\pm 0.03}$ | $23.09_{\pm 0.02}$ | $23.58_{\pm 0.04}$ |

Table 3: Mental health monitoring performance on different text embedding methods at epoch 200.

| Methods | BERT-base | RoBERTa-base | DistilBERT-base |
|---|---|---|---|
| Pretrained | 0.560 | 0.729 | 0.775 |
| + SODA | 0.757 | 0.696 | 0.701 |
| + SODA-single | 0.744 | 0.665 | 0.714 |

Table 4: Effect of fine-tuning on text embedding methods on mental health monitoring performance, measured in AUROC on *depression* task.

Testing each model design candidate from Section 3.1 for on-device training, we used Termux, a Linux Terminal emulator on Android, for unmodified PyTorch model training following previous work (Singapuram et al., 2022). We utilized RoBERTa-base (Liu et al., 2019) and a linear classifier as BERT and MLP, respectively, and LLaMa-7B (Touvron et al., 2023) model as LLM. To mimic *FedTherapist*'s user device training, we trained on a randomly generated text dataset that reflects the average word count from our study participants.

For CPU and memory measurement, we reported average per-core utilization on smartphone CPU and the max memory usage from the 5-min execution of each sub-scenarios. For latency of *speech text collection module*, we measured the time to process a 15-seconds audio file that our module processes. We measured the time to train one epoch as latency for on-device training. We used two commodity Android smartphones for the experiment: Pixel (2016) and Galaxy S21 (2021).

Table 5 presents system overhead results. There is a significant gap in CPU and memory usage between the data collection sub-scenarios. Duty cycling exhibits minimal CPU (< 1.5%) and memory usage (< 200MB), while the *speech text collection module* significantly increases CPU (> 15%) and memory (> 1GB) usage due to loading and operating multiple models like VoiceFilter (Wang et al., 2019). These results support our design to run the module only when the device is idle and charging. Moreover, the latency results confirm that our module processes audio in real-time with a processing time shorter than the audio length.

For FL, *End-to-End BERT+MLP* shows increased CPU (> 45%), memory usage (> 800MB), and latency (> 10 seconds per epoch) over *Fixed-BERT + MLP*. While *Fixed-BERT + MLP* still consumes > 25% in CPU, its per-epoch latency is extremely short (< 0.1 seconds), and the overhead in training is negligible. *LLM* measurement was not possible as the model was not loaded, requiring > 8GB memory, which exceeds the test smartphones' memory. We chose *Fixed-BERT + MLP* for *FedTherapist*, given that it achieves the highest mental health detection performance (Table 1) and the lowest smartphone overhead.

# F Discussion

## F.1 Context Features

We designed our context-aware language learning (*CALL*) based on eight context models, where the context models train on the text data collected at each context (Section 3.2). However, a combination of different contexts (e.g., texts that are collected at nighttime and at home, i.e., $T_N + L_H$) could be a viable option to improve the performance of *FedTherapist* with *CALL*. Moreover, contexts other than those we considered (time, location, motion, and application) could be used to further improve *CALL*. For the speech input, audio features such as a tone or pitch of a voice could be utilized. For the keyboard input, features such as typing speed (e.g., word per minute) could be leveraged. We leave such an investigation as future work.

## F.2 Applicability of *FedTherapist* on Users with Little Speech and Keyboard Input

As *FedTherapist* takes users' speech and keyboard input from smartphones, one might wonder about the performance of *FedTherapist* on users who produce less input; i.e., users who do not type much on smartphones or speak less than others. Figure 7 shows the test loss on users' input word count from speech and keyboard on the *depression* task. For either input type, a correlation between the word count and the test loss is not observed. The participants with less word count (< 10,000) mostly

| Scenario | Sub-Scenario | Device | CPU (%) | Memory (MB) | Latency (sec) |
|---|---|---|---|---|---|
| Data Collection (Section C) | Duty Cycling | Pixel | 1.16 | 119 | N/A[†] |
| | | Galaxy S21 | 0.29 | 191 | N/A[†] |
| | Speech Text Collection Module | Pixel | 49.21 | 1245 | 11.20 |
| | | Galaxy S21 | 15.84 | 1104 | 9.76 |
| On-Device Training | Fixed-BERT + MLP | Pixel | 27.05 | 215 | 0.08 |
| | | Galaxy S21 | 42.80 | 222 | 0.03 |
| | End-to-End BERT + MLP | Pixel | 85.59 | 842 | 13.63 |
| | | Galaxy S21 | 49.68 | 886 | 38.89 |
| | LLM[‡] | Pixel | N/A | N/A | N/A |
| | | Galaxy S21 | N/A | N/A | N/A |

Table 5: Smartphone overhead on scenarios of *FedTherapist*: CPU denotes average per-core utilization post 5-min task execution. [†]: restricted to measure individual Voice Activity Detector model execution from its API during duty cycling; [‡]: N/A measurements, as the model is not loaded on tested smartphones due to memory constraints.

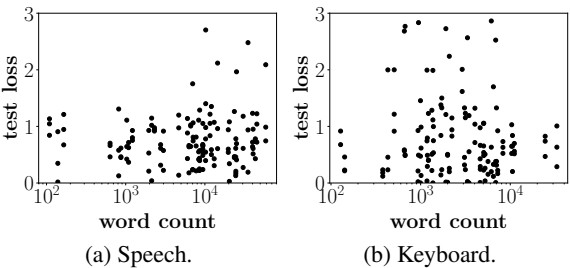

(a) Speech.    (b) Keyboard.

Figure 7: Test loss on users' input word count from speech and keyboard experiments. The results show that *FedTherapist* is applicable on users with little speech and keyboard input (< 1,000 words on each input).

resulted in low test loss (< 1). A user with only 128 words of keyboard input even resulted in a test loss of 0.154. This result shows that *FedTherapist* could also be effective on users with little speech and keyboard input.

## G   Related Work

**Mental Health Monitoring on Mobile Devices**
The early detection and treatment of mental illnesses are the keys to reducing the negative impact of the diseases (Sharp and Lipsky, 2002; Costello, 2016). Many patients are unaware of their mental health problems and this results in a delay of treatment (Lysaker et al., 2009; Epstein et al., 2010). To combat this issue, many researchers explored the possibility of passively monitoring users' mental health on mobile devices.

Most previous approaches focused on developing mental health monitoring systems with non-text data, leveraging mobile sensors and device usage statistics of a user. Wang et al. (2016, 2017, 2018) explored the association between the symptoms of schizophrenia and depression with the automati-

cally tracked smartphone-based features on sleep, mobility, conversations, and phone usage. Wang et al. (2014) also conducted the StudentLife study, which identified the correlation between smartphone sensing and the mental wellbeing measures (e.g., stress, loneliness, depression, etc.) on university students. Mehrotra et al. (2017) conducted a study that finds a correlation between users' emotional states and mobile phone interaction, that includes application usage log, notification log, and communication log (e.g., count and length of calls and SMSes). Canzian and Musolesi (2015) introduced a depression detection system based on location trajectories and daily questionnaire responses of users. DepreST-CAT (Tlachac et al., 2022a) is a depression and anxiety screening method that leverages the time-series call and text logs without the language content on smartphones. Li and Sano (2020) proposed a framework that automatically extracts efficient temporal features from skin conductance, skin temperature, and acceleration signals for mental health assessment. cStress (Hovsepian et al., 2015) is a stress monitoring system on wearable sensors that captures users' ECG (electrocardiograph) and respiration data. LiKamWa et al. (2013) and Morshed et al. (2019) developed models that predict individuals' moods based on mobile sensors or smartphone usage data.

In summary, much research has contributed to mobile mental health monitoring systems with non-text data. However, we believe that utilizing user-generated linguistic expressions could be a viable option to achieve more accurate mental health monitoring on mobile devices. Thus, we propose *FedTherapist*, a mobile system that effectively leverages text data from users to achieve

accurate mental health monitoring while preserving user privacy. While there have been few approaches that utilized the text data to perform mobile health screening on mobile devices (Liu et al., 2020; Tlachac and Rundensteiner, 2020; Tlachac et al., 2022b, 2021, 2022c), they were limited to partial text data (e.g., text data from messenger applications), or required a user's active text input. Moreover, these approaches do not discuss how such large and noisy text data could be effectively utilized for mental health prediction, nor explore different model designs for language-based mental health monitoring on smartphones. Unlike such previous approaches, *FedTherapist* fully utilizes a user's speech and keyboard input on every application on a smartphone. We explore multiple model structures, including BERT and LLMs, and propose a user context-aware methodology to effectively leverage raw text data with *FedTherapist* to achieve accurate mental health monitoring.

**Mental Health Monitoring on Social Media**
Leveraging a user's linguistic expressions for mental health screening has been widely studied for social media, where users freely express their opinions and share their status in their own words. User posts on various platforms such as Reddit (De Choudhury et al., 2016; Yates et al., 2017; Aragón et al., 2019; Ambalavanan et al., 2019), Twitter (Park et al., 2012; Coppersmith et al., 2014; Husseini Orabi et al., 2018; Chen et al., 2018), Facebook (De Choudhury et al., 2021; Saha et al., 2021), and Instagram (Reece and Danforth, 2017) have been utilized to predict the mental health status of users. Previous studies identified a specific pattern of linguistic expressions related to the mental status change or disorders; De Choudhury et al. (2016) identified distinctive markers from Reddit users related to linguistic structure and post-readability that characterize suicidal ideation. Physical behavior sensing was exploited to complement social media data in mental status prediction (Saha et al., 2021), and temporal emotion measures were leveraged to detect the physiological constructs on Twitter (Chen et al., 2018).

However, social media-based approaches have limited applicability as they can only be used by active social media users. Moreover, mental health prediction based on social media could be biased as users tend to idealize themselves on social media rather than expressing their real emotions (Vogel and Rose, 2016; Herring and Kapidzic, 2015). We focus on developing a mental health monitoring system on smartphones owned by more than six billion people (Statista, 2022). We fully leverage smartphone speech and keyboard input for effective mental health monitoring.