# OpenReview forum: "FedTherapist: Mental Health Monitoring with User-Generated Linguistic Expressions on Smartphones via Federated Learning"
_EMNLP/2023/Conference — EMNLP 2023 Main_

### Official Review · Reviewer_PGBC · 2023-08-04

**Soundness:** 3

**Excitement:**

3: Ambivalent: It has merits (e.g., it reports state-of-the-art results, the idea is nice), but there are key weaknesses (e.g., it describes incremental work), and it can significantly benefit from another round of revision. However, I won't object to accepting it if my co-reviewers champion it.

**Paper Topic And Main Contributions:**

This paper presents a system to assess the mental health of users through textual data generated by their smartphones. Textual data is extracted from keyboard and speech (microphone) inputs. Given the sensitivity of the data, authors propose to use Federated Learning, in order to preserve the privacy of the users. They also develop an ensemble architecture, which is designed to process textual data depending on the context in which it was generated. As context, authors identify Time, Location, Motion and Application, based on previous studies of the field. After collecting data from 46 users, authors compare different variants of their proposals and also a non-textual system based on previous work. They conclude that using textual information is beneficial for mental health monitoring.

**Questions For The Authors:**

A- In Section 4.1, for the method CL+NonText, you refer to a previous work, However, could you give some details of the learning mehtod? Is it a neural network? Another method?
B- For the non-experts in the area (such as myself), could you provide the meaning of the acronym IRB? It is mentioned twice in the paper, but it is not explained anywhere.

**Reasons To Accept:**

A- The field of mental health monitoring  is very important nowadays.
B- The paper is well-motivated and founded on previous work in the area of mental health.
C- The presented results are strong and support the idea of using textual data for mental health monitoring.

**Reasons To Reject:**

A- Many important details are omitted from the main body of the paper. I could find some of them in the appendices, but I think they should be in the main body, since the paper should be understood without the appendices. For example, if you don't read the appendices, you don't know that the results of Table 2 come from a leave-one-user-out cross-validation. I think this is very important. For Table 1 results, I don't still know which data splits were used and how. Another example: in Section 4, authors say they answer 4 questions. However, they also say that questions 3 and 4 are answered in Appendix E. In my opinion this should not happen. Or you answer the questions in the main body or you don't introduce the questions until the appendices.
B- In Line 162, authors mention that the lower performance of end-to-end BERT + MLP and LLM could be due to overfitting. To validate the hypothesis we would need to know train-dev-test results and learning curves. I couldn't find this information anywhere.

**Reproducibility:**

2: Would be hard pressed to reproduce the results. The contribution depends on data that are simply not available outside the author's institution or consortium; not enough details are provided.

**Reviewer Confidence:**

1: Not my area, or paper was hard for me to understand. My evaluation is just an educated guess.

---

> ### Author Rebuttal · Authors · 2023-08-29
>
> We truly appreciate your thoughtful comments and valuable suggestions. We plan to address them in our revised manuscript as below:
>
> **Comment 1: Important details being omitted from the main body**
>
> Thank you for highlighting the critical details that were inadvertently confined to the appendices. We understand and agree that a reader should be able to comprehend the essence of our research without having to cross-reference the supplementary sections.
>
> In direct response to your feedback:
>
> * We will integrate the experimental methodologies for Tables 1 and 2 into the main body, ensuring that readers can grasp our methodology without referencing the appendices.
> * We will ensure that the responses to the questions introduced in Section 4 are presented within the main body itself.
>
> The extra page permitted for the camera-ready version will greatly aid in making these revisions. We sincerely appreciate your feedback and commit to ensure the manuscript's clarity and self-completeness within the main body.
>
> **Comment 2: Validating the performance of end-to-end BERT + MLP and LLM**
>
> Thank you for your feedback on verifying the over-fitting of BERT and LLM experiments. We re-ran the experiments to share the learning curve of train-dev(validation)-test sets, which we will include in our revised manuscript. Specifically:
>
> We re-ran the experiments by performing LOUO cross-validation, splitting users into one test user, five validation users, and 40 train users. We trained until 30 epochs and five epochs for RoBERTa and LLaMa, respectively as in LoRA [1], as follows:
>
> **End-to-End BERT learning curve (Loss):**
>
> | Epochs     | 0     | 5     | 10    | 15    | 20    | 25    | 30    |
> |------------|-------|-------|-------|-------|-------|-------|-------|
> | Train loss | 0.696 | 0.634 | 0.583 | 0.528 | 0.478 | 0.437 | 0.418 |
> | Val loss   | 0.696 | 0.754 | 0.791 | 0.841 | 0.931 | 0.978 | 1.004 |
> | Test loss  | 0.702 | 0.742 | 0.801 | 0.880 | 0.999 | 0.993 | 1.030 |
>
> **End-to-End BERT learning curve (AUROC):**
>
> | Epochs     | 0     | 5     | 10    | 15    | 20    | 25    | 30    |
> |------------|-------|-------|-------|-------|-------|-------|-------|
> | Train AUROC | 0.457 | 0.392 | 0.523 | 0.406 | 0.511 | 0.637 | 0.600 |
> | Val AUROC  | 0.477 | 0.248 | 0.346 | 0.482 | 0.325 | 0.407 | 0.454 |
> | Test AUROC  | 0.438 | 0.423 | 0.411 | 0.351 | 0.253 | 0.361 | 0.401 |
>
> **LLM learning curve (Loss):**
>
> | Epochs     | 0     | 1     | 2     | 3     | 4     | 5     |
> |------------|-------|-------|-------|-------|-------|-------|
> | Train loss | 1.502 | 0.714 | 0.694 | 0.682 | 0.617 | 0.544 |
> | Val loss   | 1.285 | 0.788 | 0.724 | 0.770 | 0.767 | 0.845 |
> | Test loss  | 1.061 | 0.777 | 0.747 | 0.775 | 0.742 | 0.811 |
>
> **LLM learning curve (AUROC):**
>
> | Epochs     | 0     | 1     | 2     | 3     | 4     | 5     |
> |------------|-------|-------|-------|-------|-------|-------|
> | Train AUROC | 0.525 | 0.566 | 0.451 | 0.501 | 0.429 | 0.575 |
> | Val AUROC | 0.596 | 0.337 | 0.446 | 0.462 | 0.367 | 0.316 |
> | Test AUROC | 0.686 | 0.389 | 0.383 | 0.323 | 0.400 | 0.345 |
>
> As shown in the above results, both end-to-end BERT and LLM models show low train loss and high validation and test loss over the epochs, which shows the overfitting of those models during fine-tuning. We hypothesize the overfitting of BERT and LLM happens as our training samples are only from 40 users; previous studies [2, 3] corroborate that fine-tuning the pre-trained models can overfit and lose generalization on smaller-scale train datasets.
>
> **Comment 3 (Question A): Detailed explanation of a previous work (CL + NonText)**
>
> For CL+NonText, we followed the approach outlined by Wang et al. [4]. This involves training a linear model on the following lasso-selected mobile features: stationary time, conversation count, heart rate, sleep end time, location duration, unlock duration, unlock counts, and number of places visited. As we target a smartphone-based system, we excluded the heart rate feature, which Wang et al. measured on a wristband. We extracted the features by preprocessing the device logs and raw sensor values such as GPS, accelerometer, or ambient light sensor from participant’s mobile devices. Our revised manuscript will provide a more detailed description of this method for clarity.
>
> **Comment 4 (Question B): Meaning of the acronym IRB**
>
> Thank you for highlighting this oversight. IRB stands for “Institutional Review Board”. IRB’s approval is essential for conducting research involving human participants. The primary purpose of securing IRB approval is to guarantee the ethical treatment and protection of human subjects involved in the study [5]. In our revised manuscript, we will ensure that this acronym is clearly defined for the benefit of all readers.
>
> **References:**
>
> [1] Hu, Edward J., et al. "LoRA: Low-rank adaptation of large language models." ICLR 2022.
>
> [2] Dai, Andrew M., and Quoc V. Le. "Semi-supervised sequence learning." NeurIPS 2015.
>
> [3] Sun, Chi, et al. "How to fine-tune bert for text classification?." CCL 2019.
>
> [4] Wang, Rui, et al. "Tracking depression dynamics in college students using mobile phone and wearable sensing." ACM IMWUT 2018.
>
> [5] Morris, Marilyn C., and Jason Z. Morris. "The importance of virtue ethics in the IRB." Research Ethics 12.4 (2016): 201-216.

---

### Official Review · Reviewer_YJZA · 2023-08-05

**Soundness:** 3

**Excitement:**

4: Strong: This paper deepens the understanding of some phenomenon or lowers the barriers to an existing research direction.

**Paper Topic And Main Contributions:**

This paper presents "FedTherapist," a privacy-preserving mobile system for mental health monitoring using user-generated text data (speech and keyboard input) on smartphones. The system employs Federated Learning to decentralize model training, ensuring privacy. The authors explore multiple model designs, concluding that the Fixed-BERT + MLP model offers the best balance between performance and smartphone overhead. They also introduce the Context-Aware Language Learning (CALL) methodology, which integrates various temporal contexts of a user to enhance the model's ability to sense mental health signals. The evaluation shows that FedTherapist with CALL achieves higher accuracy in predicting mental health statuses than models trained with non-text data.




**Questions For The Authors:**

Question A: Could you provide more details on the privacy-preserving measures implemented in FedTherapist beyond the use of Federated Learning? How are potential privacy risks mitigated?

Question B: The evaluation of the system was conducted on a relatively small sample size of 46 participants. How do you plan to validate the system's effectiveness on a larger scale?

Question C: Could you elaborate on the comparative analysis with other existing methods or systems for mental health monitoring? How does FedTherapist perform in comparison to these methods?

Question D: Are there plans to release the software or pre-trained models to the community to enhance reproducibility and further research in this area?

**Reasons To Accept:**

1 The paper presents an innovative approach to mental health monitoring by leveraging user-generated text data on smartphones. This represents a significant step forward in the application of NLP techniques to real-world health monitoring.

2 The use of Federated Learning in the proposed system, FedTherapist, addresses significant privacy concerns associated with the use of personal data for health monitoring. This could set a precedent for future research in this area.

3 The empirical evaluation of the proposed system using real-world user data adds credibility to the research and provides a benchmark for future studies in this area.



**Reasons To Reject:**

1 The evaluation of the proposed system is conducted on a relatively small sample size of 46 participants. This might limit the generalizability of the results. A larger-scale study would provide more robust evidence of the system's effectiveness.

2 The paper could benefit from a more extensive comparative analysis with other existing methods or systems for mental health monitoring. This would provide a clearer picture of the relative strengths and weaknesses of the proposed system.

3 While the paper emphasizes the privacy-preserving nature of the proposed system, it does not provide a detailed analysis of potential privacy risks or how they are mitigated, beyond the use of Federated Learning. A more thorough discussion of privacy implications would strengthen the paper.

4 The paper does not explicitly mention the release of software or pre-trained models, which could limit the reproducibility of the research and its impact on the community.

**Reproducibility:**

3: Could reproduce the results with some difficulty. The settings of parameters are underspecified or subjectively determined; the training/evaluation data are not widely available.

**Reviewer Confidence:**

4: Quite sure. I tried to check the important points carefully. It's unlikely, though conceivable, that I missed something that should affect my ratings.

---

> ### Author Rebuttal · Authors · 2023-08-29
>
> We are deeply grateful for your thoughtful comments, especially for recognizing our work as an innovative approach and a significant step toward real-world mental health monitoring. We plan to address your valuable suggestions in our revised manuscript as follows:
>
> **Comment 1 (Question A): Privacy analysis beyond federated learning**
>
> Thank you for emphasizing the importance of privacy preservation in FedTherapist. Given that recent studies [1, 2, 3] have highlighted potential data leakage risks associated with sharing model updates in FL, we concur that a rigorous privacy analysis of FedTherapist is paramount.
>
> In Appendix F.2 of our submitted paper, we discussed the privacy measures adopted in FedTherapist in detail. To briefly summarize:
>
> * FedTherapist’s Fixed BERT + MLP design only trains MLP via FL, which takes BERT embeddings as an input. Such a design hinders potential attackers' direct recovery of raw user text.
> * However, we acknowledged the risk of attackers possibly recovering original text from embedded representations [4, 5]. To tackle this, we discussed integrating protection methodologies to such an attack [6] with FedTherapist as a future work.
> * Lastly, the risk exists where an attacker might deduce private user information based on non-zero gradients of specific context models. Our proposed solution is to adopt a widely used Secure Aggregation protocol [7] to prevent an attacker from acquiring individual model updates during FL.
>
> In our revised manuscript, we will include a summarized privacy analysis of FedTherapist within the main body of the paper.
>
>
> **Comment 2 (Question B): Small-scale experiment**
>
> Thank you for your comment regarding the scale of our study. Due to the sensitive nature of our collected data, including chat messages and medical status, our compensatory costs per participant ($72 for ten days) were higher than those of typical non-invasive studies. This, combined with budget limitations, led to our sample size of 46 participants. However, it's worth highlighting that our study draws from a diverse age range from the 20s to 60s, in contrast to prior smartphone-based studies that primarily targeted university students [8, 9].
>
> A large-scale study remains our paramount objective and is our most desired future work. Moving forward, we plan to deploy federated learning on user smartphones, eliminating the need for direct data collection, which would be cost-effective and allow us to validate our findings at a larger scale.
>
> **Comment 3 (Question C): Comparative analysis with other existing methods**
>
> Thank you for your feedback. We conducted a comprehensive comparative analysis of FedTherapist against closely related approaches:
>
> * **CNN + MLP:** Modeled after Yates et al. [10] and Kim et al. [11], this CNN-based text classification employs a convolutional layer and an MLP (Multi-Layer Perceptron). We used Yates et al. 's hyperparameters for depression identification.
> * **Bag-of-words + MLP:** This model employs an MLP with a bag-of-words input with TF-IDF, based on Pirina et al.'s [12] depression detection. We experimented with a n-gram range of [1, 3] and representation sizes from [1000, 5000, 10000, 20000].
> * **Empath + MLP:** Taking cues from Tlachac et al. [13], which utilizes smartphone SMS messages, the model utilizes output representations from Empath [14], which analyzes text over 200 lexical categories.
>
> To ensure a fair comparison, we trained all baseline models using learning rates from [0.0001, 0.001, 0.01, 0.1] and evaluated their performance at each of the 1000 epochs. We presented the best performance achieved by these models over the 1000 epochs in the table below. For FedTherapist, we reported the test AUROC after its 1000 epochs without selecting its peak performance. Despite this, FedTherapist still demonstrated superior results as follows:
>
>
> | Methods            | FedTherapist | CNN+MLP | Bag-of-words+MLP | Empath+MLP |
> |--------------------|--------------|---------|------------------|------------|
> | Depression (AUROC) | 0.721        | 0.418   | 0.564            | 0.635      |
>
> The result shows that FedTherapist outperforms other methods in depression detection on our dataset. Limited training samples from only 46 users in our dataset might have hindered the effectiveness of CNNs trained from scratch — in contrast, Yates et al. trained with Reddit posts from more than 100,000 users. Other results suggest that using frozen BERT for text representations in FedTherapist shows better mental health monitoring performance than using bag-of-words or Empath.
>
> In our revised manuscript, we will clarify the points above and include results from the regression mental health monitoring tasks. We anticipate these to exhibit a trend similar to that in depression.
>
> **Comment 4 (Question D): Plans to release the software or pre-trained models**
>
> Thank you for your suggestion. Following the publication of our paper, we intend to release both (1) the Android application utilized during our data collection phase and (2) the source code for data preprocessing and model training (including federated learning). This will empower the research community to replicate and build upon our work. However, in line with our approved IRB (Institutional Research Board) proposal, we cannot share the collected or processed user data and the pre-trained models to protect participants' privacy.
>
> **References:**
>
> [1] Zhu, Ligeng, Zhijian Liu, and Song Han. "Deep leakage from gradients." NeurIPS 2019.
>
> [2] Geiping, Jonas, et al. "Inverting gradients-how easy is it to break privacy in federated learning?." NeurIPS 2020.
>
> [3] Huang, Yangsibo, et al. "Evaluating gradient inversion attacks and defenses in federated learning." NeurIPS 2021.
>
> [4] Coavoux, Maximin, et al. “Privacy-preserving Neural Representations of Text” EMNLP 2018.
>
> [5] Song, Congzheng, and Ananth Raghunathan. "Information leakage in embedding models." ACM CCS 2020.
>
> [6] Lee, Garam, et al. “Privacy-Preserving Text Classification on BERT Embeddings with Homomorphic Encryption.” NAACL 2022.
>
> [7] Bonawitz, Keith, et al. "Practical secure aggregation for privacy-preserving machine learning." ACM CCS 2017.
>
> [8] Wang, Rui, et al. "StudentLife: assessing mental health, academic performance and behavioral trends of college students using smartphones." ACM UbiComp 2014.
>
> [9] Li, Boning, and Akane Sano. "Extraction and interpretation of deep autoencoder-based temporal features from wearables for forecasting personalized mood, health, and stress." ACM IMWUT 2020.
>
> [10] Yates, Andrew, Arman Cohan, and Nazli Goharian. "Depression and self-harm risk assessment in online forums." EMNLP 2017.
>
> [11] Kim, Yoon. "Convolutional neural networks for sentence classification." EMNLP 2014.
>
> [12] Pirina, Inna, and Çağrı Çöltekin. "Identifying depression on reddit: The effect of training data." EMNLP Workshop SMM4H 2018.
>
> [13] Tlachac, M. L., Katherine Dixon-Gordon, and Elke Rundensteiner. "Screening for suicidal ideation with text messages." IEEE BHI 2021.
>
> [14] Fast, Ethan, Binbin Chen, and Michael S. Bernstein. "Empath: Understanding topic signals in large-scale text." ACM CHI 2016.

---

### Official Review · Reviewer_Y3PY · 2023-08-11

**Soundness:** 3

**Excitement:**

3: Ambivalent: It has merits (e.g., it reports state-of-the-art results, the idea is nice), but there are key weaknesses (e.g., it describes incremental work), and it can significantly benefit from another round of revision. However, I won't object to accepting it if my co-reviewers champion it.

**Missing References:**

Matero, M., Idnani, A., Son, Y., Giorgi, S., Vu, H., Zamani, M., Limbachiya, P., Guntuku, S.C. and Schwartz, H.A., 2019, June. Suicide risk assessment with multi-level dual-context language and BERT. In Proceedings of the sixth workshop on computational linguistics and clinical psychology (pp. 39-44).

Liu, T., Liang, P.P., Muszynski, M., Ishii, R., Brent, D., Auerbach, R., Allen, N. and Morency, L.P., 2020. Multimodal privacy-preserving mood prediction from mobile data: A preliminary study. arXiv preprint arXiv:2012.02359.

**Paper Topic And Main Contributions:**

FedTherapist is introduced in this work which is a federated learning method that can be used on mobile devices for tracking mental health of the device user with their spoken and written communication.  They further explore a context aware method by adding contexual features like time of the day, to capture user's emotional state which shows improvement in results. The federated learning method allows user privacy by not having to transfer user data out of device, with on-device models updates with user data.

**Questions For The Authors:**

- What procedure was used for hyperparameter tuning for end-to-end BERT since the claim is that the model is over-fitting. Was a validation set used for it or some early stopping used instead of using 200 epochs as mentioned in appendix?

**Reasons To Accept:**

- This work is of interest not just to NLP community but also to broader mental health community, with an attempt to provide a solution to an important problem
- Paper is well written and easy to follow
- Authors made a huge effort from data collection to a technical solution for the problem which can use both speech / text data from keyboard input, which pushes the area of mental health and therapy forward


**Reasons To Reject:**

- Not clear if hyper-parameter tuning was done and how it was done, as they reported results for end-to-end BERT are lower than a frozen BERT model which is contradiction to similar papers (Matero, 2019)
- Method is not particularly novel given similar approaches have been proposed on multi-modal data (Liu, 2020) for mental health before, though use of speech + text is novel

Matero, M., Idnani, A., Son, Y., Giorgi, S., Vu, H., Zamani, M., Limbachiya, P., Guntuku, S.C. and Schwartz, H.A., 2019, June. Suicide risk assessment with multi-level dual-context language and BERT. In Proceedings of the sixth workshop on computational linguistics and clinical psychology (pp. 39-44).

Liu, T., Liang, P.P., Muszynski, M., Ishii, R., Brent, D., Auerbach, R., Allen, N. and Morency, L.P., 2020. Multimodal privacy-preserving mood prediction from mobile data: A preliminary study. arXiv preprint arXiv:2012.02359.

**Reproducibility:**

2: Would be hard pressed to reproduce the results. The contribution depends on data that are simply not available outside the author's institution or consortium; not enough details are provided.

**Reviewer Confidence:**

3: Pretty sure, but there's a chance I missed something. Although I have a good feel for this area in general, I did not carefully check the paper's details, e.g., the math, experimental design, or novelty.

---

> ### Author Rebuttal · Authors · 2023-08-29
>
> We greatly appreciate your thoughtful comments and feedback. In our revised manuscript, we intend to address these points as follows:
>
> **Comment 1 (Question A): Hyperparameter tuning procedure for end-to-end BERT**
>
> We appreciate your question regarding the hyperparameter tuning process for our end-to-end BERT experiments. In the updated manuscript, we will provide a detailed clarification of this procedure, outlined below:
>
> For the *previous end-to-end BERT experiment* in Section 3.1, we applied LoRA [1], a lightweight fine-tuning technique primarily based on hyperparameters from Huggingface's RoBERTa fine-tuning example [2]. Specifically:
>
> * We selected from a learning rate set of [0.0005, 0.0004, 0.0003, 0.0002, 0.00003], as recommended in the LoRA paper, reporting the highest AUROC. We used the linear learning rate decay scheduler as in LoRA.
> * While the LoRA paper for RoBERTa used a batch size of 16 or 32, we only used 16 due to our GPU memory restrictions (24GB on NVIDIA RTX 3090).
> * We performed cross-validation on 46 clients with the Leave-One-User-Out (LOUO) strategy by splitting the data into 1 test user and 45 train users. We trained until 30 epochs as in LoRA, and tested AUROC for every epoch, repeating such a process 46 times by setting each user as a test user. In the paper, we reported the highest average AUROC from all 30 epochs. Note that for Fixed-BERT + MLP (FedTherapist), we straightforwardly reported the test AUROC after its 1000 epochs without selecting its peak performance.
>
> To enhance clarity, we conducted a *new experiment* using LOUO cross-validation, segmenting the users into 1 test user, 5 for validation, and 40 for training. LLaMa 7B was also experimented as LLM, with parameters based on LoRA’s GPT-2 tuning. The table below presents the average test AUROC at the epoch with the lowest validation loss:
>
> | Methods| Fixed-BERT + MLP | End-to-End BERT + MLP | LLM   |
> |-----|-------|------|-------|
> | Depression (AUROC)| 0.716            | 0.524                 | 0.406 |
>
> Fixed-BERT+MLP outperforms End-to-End BERT+MLP and LLM in experiments that included validation set users. This advantage results from BERT and LLM overfitting on our 40-user training sample. Past research [3, 4] corroborates that pre-trained models can overfit on small-scale datasets, compromising their generalizability. This disparity with other studies stems from our dataset size, and we expect our results to align with others when applying a larger, more diverse training dataset.
>
> **Comment 2: Novelty compared to a previous approach**
>
> We sincerely appreciate that you highlighted a closely related work. We would like to clarify FedTherapist’s novelty compared with a previous approach (Liu et al., 2020 [5]) as follows:
>
> * **Multi-Context Integration:** FedTherapist involves Context-Aware Language Learning (CALL), which integrates four types of smartphone contexts — time, location, motion, and app — to enhance the model’s ability to sense mental health signals. In contrast, Liu et al.’s approach solely incorporates one type of context, the app usage, with text data.
> * **Modeling Technique:** FedTherapist employs ensemble learning across multiple context models to aid accurate mental health sensing with CALL. It differs from Liu et al.’s approach, which directly concatenates the text and the app usage features as an input of a single model.
> * **Model Design Exploration:** FedTherapist explores candidate model designs for language-based mental health monitoring on smartphones. This includes comparing BERT and LLM models' performance and smartphone overhead (CPU and memory usage). Conversely, Liu et al. solely utilize a bag-of-words methodology. For a more comprehensive comparison, we plan to add the bag-of-words-based model design in our comparison in our revised manuscript.
> * **Data Types:** We've designed and evaluated FedTherapist on two types of user text data from smartphones: speech and keyboard input, while Liu et al. only leveraged the keyboard input.
>
> In our revised manuscript, we will clarify our novelty compared to Liu et al.’s approach.
>
> **References:**
>
> [1] Hu, Edward J., et al. "LoRA: Low-rank adaptation of large language models." ICLR 2022.
>
> [2] HuggingFace, “LoRA Sequence Classification Example”, https://github.com/huggingface/peft/blob/main/examples/sequence_classification/LoRA.ipynb
>
> [3] Dai, Andrew M., and Quoc V. Le. "Semi-supervised sequence learning." NeurIPS 2015.
>
> [4] Sun, Chi, et al. "How to fine-tune bert for text classification?." CCL 2019.
>
> [5] Liu, Terrance, et al. “Multimodal privacy-preserving mood prediction from mobile data: A preliminary study.” arXiv preprint arXiv:2012.02359.

---

### Meta-Review · Area_Chair_SGjC · 2023-09-17

**Recommendation:** 5

**Metareview:**

In general, reviewers found that the paper sufficiently supported its major claims and arguments, with only minor points requiring extra support or details.  Their enthusiasm ranged from ambivalent to strong, with Reviewer YJZA in particular advocating for the paper and noting that the work presents an innovative path forward in the application of NLP to real-world health monitoring.

**Summary of Reviewer Feedback and Discussion:**
- **Reviewer Y3PY** liked that the work is broadly impactful and describes a substantial effort to collect data and develop a comprehensive technical solution.  However, they were uncertain about some aspects of the paper, such as whether hyperparameters were tuned, and they felt that the proposed method itself was only novel in its use of speech and text as modalities.  In their rebuttal, the authors provided clarifying information regarding their hyperparameter tuning process, as well as an additional experiment to enhance clarity.  They also more fully elaborated on FedTherapist's novelty compared to the prior approach mentioned by Reviewer Y3PY.  They promised to incorporate these revisions in the updated manuscript.  Reviewer Y3PY thanked the authors for the detailed rebuttal, and noted that while their concerns regarding hyperparameter tuning were addressed, they were still unconvinced of the approach's novelty.
- **Reviewer YJZA** felt that the work was innovative and addressed important privacy concerns associated with the use of personal data for health monitoring.  They also appreciated the empirical evaluation of the proposed system.  However, they noted that the evaluation was conducted on a relatively small sample size and that the work could benefit from more extensive comparison with other systems for mental health monitoring.  They also wished that the paper had included a detailed analysis of potential privacy risks and how the proposed approach mitigates them, and they were curious whether the software or pretrained models would be released publicly.  In their rebuttal, the authors noted that some privacy analysis is provided in the appendix but that in their revised manuscript they will summarize this in the main body of the paper.  They also acknowledged the limited sample size of their study, explaining that it was due to budget limitations and the sensitive nature of their work, but noted that they did ensure that their participant pool was distributed across a broader demographic group than seen in prior work.  They included some additional results comparing their approach to existing methods, and they noted that while they plan to release the application used for data collection and the source code for data preprocessing and model training, in keeping with IRB protocol they cannot share their data or pretrained models to protect participants' privacy.
- **Reviewer PGBC** liked that the work tackled an important and well-motivated problem, and that the presented results are strong and support the idea of using text data for mental health monitoring.  However, they felt that many important details were omitted from the main body of the paper or relegated to the appendix.  They also asked some questions pertaining to specific methodological details.  In their rebuttal, the authors noted specific elements from the appendix that they plan to move to the main body of the paper, given additional space following acceptance.  They also ran additional experiments based on questions asked by Reviewer PGBC, and presented these new results to further support the work.  They provided clarifying responses to the reviewer's other questions.

---

### Decision · Program_Chairs · 2023-10-07

**Decision:**

Accept-Main

**Comment:**

In general, reviewers found that the paper sufficiently supported its major claims and arguments, with only minor points requiring extra support or details.  Their enthusiasm ranged from ambivalent to strong, with Reviewer YJZA in particular advocating for the paper and noting that the work presents an innovative path forward in the application of NLP to real-world health monitoring.

**Summary of Reviewer Feedback and Discussion:**
- **Reviewer Y3PY** liked that the work is broadly impactful and describes a substantial effort to collect data and develop a comprehensive technical solution.  However, they were uncertain about some aspects of the paper, such as whether hyperparameters were tuned, and they felt that the proposed method itself was only novel in its use of speech and text as modalities.  In their rebuttal, the authors provided clarifying information regarding their hyperparameter tuning process, as well as an additional experiment to enhance clarity.  They also more fully elaborated on FedTherapist's novelty compared to the prior approach mentioned by Reviewer Y3PY.  They promised to incorporate these revisions in the updated manuscript.  Reviewer Y3PY thanked the authors for the detailed rebuttal, and noted that while their concerns regarding hyperparameter tuning were addressed, they were still unconvinced of the approach's novelty.
- **Reviewer YJZA** felt that the work was innovative and addressed important privacy concerns associated with the use of personal data for health monitoring.  They also appreciated the empirical evaluation of the proposed system.  However, they noted that the evaluation was conducted on a relatively small sample size and that the work could benefit from more extensive comparison with other systems for mental health monitoring.  They also wished that the paper had included a detailed analysis of potential privacy risks and how the proposed approach mitigates them, and they were curious whether the software or pretrained models would be released publicly.  In their rebuttal, the authors noted that some privacy analysis is provided in the appendix but that in their revised manuscript they will summarize this in the main body of the paper.  They also acknowledged the limited sample size of their study, explaining that it was due to budget limitations and the sensitive nature of their work, but noted that they did ensure that their participant pool was distributed across a broader demographic group than seen in prior work.  They included some additional results comparing their approach to existing methods, and they noted that while they plan to release the application used for data collection and the source code for data preprocessing and model training, in keeping with IRB protocol they cannot share their data or pretrained models to protect participants' privacy.
- **Reviewer PGBC** liked that the work tackled an important and well-motivated problem, and that the presented results are strong and support the idea of using text data for mental health monitoring.  However, they felt that many important details were omitted from the main body of the paper or relegated to the appendix.  They also asked some questions pertaining to specific methodological details.  In their rebuttal, the authors noted specific elements from the appendix that they plan to move to the main body of the paper, given additional space following acceptance.  They also ran additional experiments based on questions asked by Reviewer PGBC, and presented these new results to further support the work.  They provided clarifying responses to the reviewer's other questions.